# Protocol for the process evaluation of the Promoting Activity, Independence and Stability in Early Dementia (PrAISED), following changes required by the COVID-19 pandemic

Claudio Di Lorito ,[1] Alessandro Bosco,[2] Sarah E Goldberg,[3] Roshan Nair,[2] Rebecca O'Brien,[1] Louise Howe,[1] Veronika van der Wardt,[4] Kristian Pollock,[3] Vicky Booth,[1] Pip Logan,[1] Maureen Godfrey,[1] Marianne Dunlop,[1] Jane Horne,[1] Rowan H Harwood[3]

[1]Division of Rehabilitation, Ageing and Wellbeing, University of Nottingham, Nottingham, UK
[2]Division of Psychiatry and Applied Psychology, University of Nottingham, Nottingham, UK
[3]School of Health Sciences, University of Nottingham, Nottingham, UK
[4]Philipps-Universitat Marburg, Marburg, Hessen, Germany

**Correspondence to**
Dr Claudio Di Lorito;
claudio.dilorito@nottingham.ac.uk

## ABSTRACT

**Introduction** The Promoting Activity, Independence and Stability in Early Dementia (PrAISED) randomised controlled trial (RCT) is evaluating a home-based, face-to-face, individually tailored, activity and exercise programme for people living with dementia. Social distancing requirements following the COVID-19 pandemic necessitated rapid changes to intervention delivery. **Methods and analysis** A mixed-methods process evaluation will investigate how the changes were implemented and the impact that these have on participants' experience. An *implementation study* will investigate how the intervention was delivered during the pandemic. A *study on the mechanisms of impact and context* will investigate how these changes were experienced by the PrAISED participants, their carers and the therapists delivering the intervention. The study will commence in May 2020.

**Ethics and dissemination** The PrAISED RCT and process evaluation have received ethical approval number 18/YH/0059. The PrAISED process evaluation will enable us to understand how distancing and isolation affected participants, their activity and exercise routines and whether the therapy programme could be continued with remote support. This will be valuable both in explaining trial results and also contribute to understanding and designing new ways of delivering home-based services and rehabilitation interventions for people with dementia and their carers.

**Trial registration number** ISRCTN15320670; Pre-results.

## INTRODUCTION

Dementia is a neurodegenerative condition characterised by a cluster of symptoms, including memory loss and deterioration of motor skills.[1–4] More than 50 million people in the world live with dementia.[5] Projections estimate that this number will rise to 130 million people in the next 30 years.[5] Dementia presents enormous financial

### Strengths and limitations of this study

► This study will capture the full range of perspectives, by involving in research participants with dementia, their carers and professionals delivering the intervention.
► This study will gather a holistic picture of the phenomenon, as it uses different methodologies, including quantitative and qualitative data and data triangulation.
► This study will collect qualitative data at two time points, to capture progress over time.
► The qualitative interviews in this study will be carried out remotely, which could pose barriers to participants with dementia.
► This process evaluation team is not independent of the main trial team and this may generate confirmation bias of study hypotheses.

burden.[6] In the UK alone, the cost of health and social care for people with the condition is £50 billion, which will grow to £140 billion by 2040.[5] Keeping physically active has benefits for people with dementia on executive functioning, mobility, activities of daily living, independence and quality of life (QoL),[7–22] which have been linked to a reduced risk of falls, hospital admissions and health and social care costs.

A number of physical activity and exercise intervention programmes have been developed for people with dementia.[15 16] Among these is the Promoting Activity, Independence and Stability in Early Dementia (PrAISED),[23] an intervention to promote activity and independence in people with early dementia or mild cognitive impairment, whose clinical and cost-effectiveness is being evaluated in a

five-site randomised controlled trial (RCT). So far, out of a total recruitment target of 368 participants, 300 participants have been randomised to either a control group (receiving brief falls assessment and advice only) or an intervention arm.[24] Participants in the intervention arm receive an individually tailored programme of up to 50 visits at home over a period of 52 weeks from a multidisciplinary team, including physiotherapists (PTs), occupational therapists (OTs) and rehabilitation support workers (RSWs).[24] The PrAISED programme comprises: physical exercises (ie, progressive strength, balance and dual-task); functional activities (ie, activities of daily living with an element of physical activity, such as going out for food shopping); promotion of inclusion in community life (eg, through provision of information on physical exercise group classes); risk enablement (ie, assessing, mitigating and agreeing on risks to be taken or avoided) and environmental assessment.[24]

The PrAISED RCT includes a process evaluation,[25] which aims to describe and quantify intervention delivery, identify the key elements that make the intervention effective and the variables affecting participants motivation to adhere to the programme and remain physically active in the long-term (ie, beyond the active intervention period). These variables, which have been recently synthesised in a theoretical model,[26 27] include the social opportunities linked to exercise, the therapeutic relationship built with the therapists delivering the intervention, family or carer support, the availability and inclusion of the person in community (physical) activities, the accessibility of the environment (eg, availability of parks, public transport) and the notion of independence and autonomy (eg, how, when and where to exercise).

In March 2020, many of the elements enabling and supporting participants in the PrAISED programme became impossible to deliver due to the pandemic of COVID-19. Measures to slow the spread of the virus were advised and then mandated by governments.[28–30] People over 70 years of age, especially those with pre-existing conditions, were told to self-isolate to shield them from increased risk of illness, complications, hospitalisation and mortality.[31 32]

The negative effects that social isolation may have on the health and well-being of older people are well known.[33] In people with dementia, there might be additional effects, such as a negative impact on functioning, through loss of opportunity to engage with family or in activities outside the home. In order to continue the trial and maintain an element of social contact during this unprecedented time, changes were made to the PrAISED programme intervention delivery (table 1). There were no changes in training, as all therapists delivering PrAISED had been recruited and trained before the amendment to PrAISED. Instead, the therapists were provided with new written guidance on how to deliver the intervention remotely (online supplementary appendix 1). The participants who were still receiving the intervention when these changes occurred (March 2020) (n=213) automatically started receiving the amended version of the PrAISED programme. The main change was that participants would not receive visit from therapists at home, as this would place them at risk of contracting the virus. Instead, the therapists would continue to support the participants remotely, by telephone or video, in line with the Chartered Society of Physiotherapists guidance.[34]

These changes might have important implications on the participants' experience of the intervention. Previous studies have found that face-to-face support from therapists facilitates the creation of a strong therapeutic alliance with the person with dementia, which proves an effective tool for adherence.[27] Home visits may facilitate coproduction of a programme tailored to the person's needs and aspirations, which is linked to feelings of empowerment and autonomy.[35] They may also prove positive for the carers, who, as a result of their caring duties, may risk social isolation[36 37] and reduced QoL.[38] On the other hand, face-to-face support can increase feelings of dependency among participants, potentially resulting in separation anxiety towards the end of the intervention period.[27] From the therapists' perspective, delivering an intervention in the participants' homes can be time-consuming. It has been reported in previous process evaluations that adding travelling times on top of the existing workload might thwart job satisfaction.[39] The use of remote support might rectify some of these negative experiences.

We aim to extend the process evaluation of the PrAISED,[25] to investigate the impact of the changes made to PrAISED. Specifically, the proposed study will respond to the research questions:

**Table 1** Main changes made to the Promoting Activity, Independence and Stability in Early Dementia intervention, compared with the original version[23 24]

| Delivery of intervention | Provision of support to the therapists |
| --- | --- |
| The therapists were provided with written guidance on how to deliver the intervention (online supplementary appendix 1) | Increased access to:<br>▶ Monthly teleconferences across all sites.<br>▶ Teleconferences at individual sites.<br>▶ Provision of a regularly updated list of resources.<br>▶ Provision of informal support through email and phone.<br>▶ Provision of information and support tailored to the situation and change in practice. |

- ► How does staying at home, with no current possibility of receiving face-to-face support from therapists, affect the uptake and retention of a physical activity and exercise programme in participants with dementia? How does it affect their ability to remain independent and their QoL? Are there ways in which people with dementia can be better supported to remain physically active and independent in these circumstances?
- ► How are therapists supported to deliver a physical activity and exercise programme remotely to participants with dementia? How does this support affect their confidence and ability to deliver the intervention? Are there ways in which therapists can be better supported to deliver the intervention remotely?

## METHODS AND ANALYSIS

Based on the assumption that 'if intervention X (ie, PrAISED) is delivered, the mediating variable(s) (eg, staying at home, support from therapists available only remotely) affects the way in which outcome Y (eg, uptake and retention of a physical activity and exercise) will occur', a process evaluation aims to understand how an intervention works.[40] It does so by studying the 'implementation of the intervention' (eg, how the intervention is delivered), the 'mechanisms of impact' (eg, how participants respond to the intervention being delivered) and the 'context' (eg, the physical and social environment affecting participants' response to the intervention).[40]

This process evaluation will adopt a mixed-methods approach, including quantitative data and data ensuing from qualitative interviews. It will consist of two studies: an implementation study and a study on mechanisms of impact and context (figure 1). The study will commence in May 2020 and the final results are expected to be available in May 2021.

### Patient and public involvement

The process evaluation study team includes two patient and public involvement (PPI) contributors (MG and MD), who have been involved in the development of the process evaluation and its protocol (also acting as co-authors). The PPI contributors co-designed with the main researcher (CDL) the topic guide for the qualitative interviews with participants with dementia and their carers (see details in 'Study on mechanisms of impact and context—data collection' section) and will be involved as co-raters in the qualitative analysis of the transcripts of the interviews (see details in 'Study of mechanisms of impact and context—data analysis' section) and in disseminating research findings (eg, through attending conferences, public dissemination events and co-authoring results' papers).

### Implementation study

The study on implementation will investigate how the PrAISED intervention is delivered, following changes in procedure in response to the COVID-19 pandemic. It will focus on four domains (table 2):

- ► Fidelity (ie, the consistency of delivery of PrAISED with the amended protocol).
- ► Adaptations (ie, alterations made to delivery of PrAISED to achieve better contextual fit).
- ► Dose (ie, how much PrAISED intervention is delivered).
- ► Reach (ie, the number of therapists trained to deliver PrAISED and of participants who receive the intervention).

### Participants

The implementation study will include participants with dementia in the intervention group, their carers and the

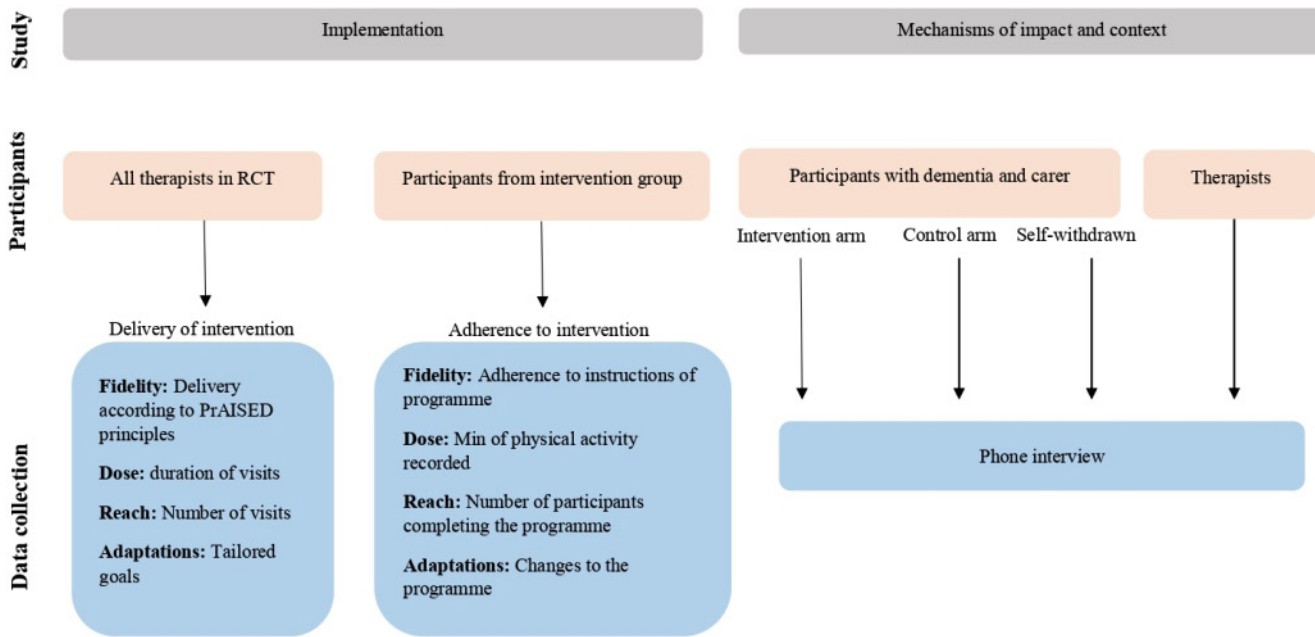

**Figure 1** Method of process evaluation.

**Table 2** Implementation study

|  | Delivery (therapists) | Adherence (participants) |
|---|---|---|
| Fidelity | Delivery of intervention against PrAISED two principles (through audio content) | Adherence to intervention as per instructions (through interview) |
| Dose | Frequency and length of contact sessions with participant* | Minutes per week recorded on calendar* |
| Reach | Number of contact sessions with participant* | Number of participants who completed the programme* |
| Adaptations | Adaptations made to deliver the sessions (through interview) | Adaptations that participants made to physical activity and exercise (through interview) |

*Data gathered during the main trial.
PrAISED, Promoting Activity, Independence and Stability in Early Dementia.

therapists who are involved in the PrAISED main trial at the time of recruitment (May 2020).

## Data collection

From the participants with dementia:

► Adherence to intervention as per instructions (fidelity), investigated through qualitative interviewing.
► Adherence to advised activity levels (dose), investigated through minutes of PrAISED activity per week as recorded on a self-completed monthly calendar (or carer-completed monthly calendar).
► The extent to which the participants with dementia come into contact with the intervention (reach), investigated by totalling the number of participants who completed the programme.
► Alterations that the participants made to achieve better contextual fit (adaptations), investigated through qualitative interviewing.

From the therapists:

Evaluation of the delivery of the adapted intervention, including:

► Number and length of remote sessions the therapists have with participants (dose and reach): A record of the date, length in minutes and therapist type (PT, OT and RSW) will be recorded for each contact. The information will be collated by the research team each week.
► Goals set for participants (adaptations): Goals that have been set with the participants will be documented by the therapists and collated centrally by the research team.
► Intervention content (fidelity, adaptations): One intervention session provided remotely by each therapist will be audio-recorded. To ensure safe handling and storing of sensitive data, the session between the therapist and the participant will be recorded remotely by one researcher within the PrAISED team with an encrypted digital audio recorder.

## Data analysis

The data from the implementation study will be analysed using IBM SPSS Statistics V.26.[41] Descriptive statistical analysis will be used to measure fidelity, dose and reach.

The audio recordings will be transferred onto an encrypted and password protected university computer server. The content will be assessed independently by two raters against 14 core principles set out in the PrAISED therapists' training manual (ie, 'visit following core principle', 'visit not following core principle' and 'principle not applicable'). An audio-analysis template will list the core principles, provide operational definitions of each of them, accompanied with practical examples of the application of principle, to facilitate retrieval of content during analysis (online supplementary appendix 2).

Prior to independent audio analysis, the two raters will pilot-test the rating procedure using a sample audio recording, to check inter-rater reliability. Scores from the two raters will be compared to determine inter-rater reliability, and if inconsistency arises in scoring, consensus will be reached through discussion between the two raters or through involvement of a third rater.

## Study on mechanisms of impact and context

The study on mechanisms of impact and context will investigate the participants and therapists' experience of the intervention, and any variable mediating intervention outcomes (eg, social distancing).

### Participants

For each research site, we will include:

1. Participants with dementia and their carer, further divided into
   – Intervention arm (ie, receiving the active intervention).
   – Control arm (ie, receiving treatment, as usual, included to investigate whether there are any relevant differences between control and intervention arm).
   – Those who have withdrawn from the therapy programme, if they agree to be interviewed.

   Purposive sampling will be carried out to ensure a diverse and representative sample in relation to gender, ethnicity, residence status (ie, living independently or living with carer) and the different research sites involved in PrAISED (ie, Nottinghamshire, Derbyshire, Lincolnshire, Somerset and Oxfordshire). The main researcher (CDL) will access

**Table 3** Conceptual Depth Scale[43]

| Criteria (with sources of evidence) | Low | Medium | High |
|---|---|---|---|
| Range (eg, frequency and variety of codes; multiplicity of data sources) | Few examples to support concepts. Only a single data-type | ⟶ | Abundant examples to support concepts. Multiple data-types |
| Complexity (eg, coding trees; positional maps; matrices) | Descriptive codes; simple or basic connections between codes; low-level analysis | ⟶ | Sophisticated networks; abstract conceptual categories which synthesise a range of codes and concepts |
| Subtlety (eg, memos; social worlds diagrams) | Conceptual language is regarded as unproblematic and one-dimensional | ⟶ | Conceptual language is understood as rich, ambiguous and multidimensional |
| Resonance (literature) | Weak resonance; emerging theory is remote from existing literature and theoretical frameworks | ⟶ | Strong resonance; emerging theory makes sense along-side existing literature; there are correlations with other theoretical frameworks, although with variations and novel-ties |
| Validity (eg, applicability test) | Low-level theorising and inward-facing; the findings have limited application to the research participants or those familiar with similar contexts. | ⟶ | Abstract level theorising and outward-facing; the findings make sense to those in the social context of the research, or ones broadly similar. |

the PrAISED RCT database and select participants from the different subgroups.

We will not exclude participants who do not have mental capacity to agree to participate or who show fluctuating capacity at the point of the interview, for the following reasons: first, they might still provide precious insight into the mechanisms of the intervention; second, their (fluctuating) cognition may have an impact and affects their response towards the intervention; finally, from an ethical standpoint, we aim to give voice to all those whose life is primarily affected by our research. However, we will take into account capacity to give consent (or lack thereof) during the course of the interview, by relying, for example, on different degrees of carer support during the session.

2. Therapists will be purposively sampled to be involved in the process evaluation. The main researcher (CDL) will access the PrAISED RCT database and select therapists from the different professions (ie, PTs, OTs and RSWs) and research sites.

In line with Guest et al,[42] we argue that, given the lack of guidance around reaching data saturation, there is a need to adopt appropriate 'tests of adequacy' for sample sizes in qualitative research. Based on the notion of 'conceptual density' (ie, gathering data until a *sufficient depth* of understanding of the domains under investigation is reached),[43] we will adopt a Conceptual Depth Scale developed by Nelson[43] (table 3), which assigns a score ranging from 1 (low) to 3 (high) to establish whether conceptual density is reached in relation to:

► 'Range' (eg, extent of diversity of data sources).

► 'Complexity' (eg, extent of networks/links across data).

► 'Subtlety' (eg, extent of similarity across data).

► 'Validity' (eg, extent to which data are transferable to other settings).

The scoring will be performed by two researchers independently of each other. The scale is used as instrument to check whether consensus is reached among researchers with respect to data saturation, rather than as quantitative assessment to determine a saturation point for data interpretation. We anticipate that conceptual density will be reached by inclusion of up to 20 participants with dementia (and 20 carers), and 20 therapists across all research sites.

### Data collection

The investigation of the mechanisms of impact and context will be based on qualitative interviews with participants. The first interview will be conducted 1 month following the change of intervention in response to the COVID-19 pandemic (ie, May 2020). Follow-up interviews will be considered, if the measures imposed following the COVID-19 pandemic are still in place, to monitor progress over time.

The interviews will consist of:

► Remote interviews (different options will be offered, including telephone or video call, depending on participants' preference) with participants with dementia and their carers (as a dyad, so that the carer can provide information, as well as support, if needed). We will use a speakerphone (for phone

interviews) for everyone to be able to contribute. Prior to the session, the researcher will mail (or email) a copy of the consent form. A verbal consent for both the participant with dementia and the carer will be recorded on tape, before the interview begins.

► Remote (phone) interviews with therapists (ie, OT, PT and RSW). Verbal consent will be recorded on tape prior to the interview.

The topic guide for the qualitative interviews is informed by the *PHYT in dementia* (PHYsical activity behaviour change Theory in dementia), whose development and validation we reported elsewhere.[26 27] Through this theoretical framework, we identified potential variables mediating intervention outcomes and developed several prompts to stimulate discussion. Exploration of context will include the impact of isolation, and its effects on exercise, activity and mental well-being.

We developed the topic guides as a collaborative effort between the research team and the PPI contributors, who helped to ensure that the interview prompts are relevant, meaningful and accessible for the participants. Although questions are study-specific, the prompts are broad in scope, to ensure that the participants feel free to express their ideas around unanticipated causal processes and consequences. The participants may also raise additional topics and issues which they feel are particularly relevant in the context of the COVID-19 pandemic, and these will be explored accordingly.

The qualitative interviews are expected to last around 40 min, depending on participants' engagement in the process, their cognitive abilities and logistics.

### Data analysis

Data will be analysed through framework analysis.[44] This method is ideal in social and healthcare qualitative research studies with large data sets. Framework analysis will ensure in-depth exploration of data, a transparent audit trail of the process of analysis and the understanding of data interpretation (eg, a description of how data link to each other and according to the objective of the study) through visual mapping.[44]

Data analysis will follow the steps for good practice in Framework Analysis identified by Gale *et al*[44]:

1. *Verbatim transcription* of the interviews by a professional transcriber, who will also anonymise data. Large margins and double line spacing in the transcripts will be left to create room for coding and note-taking.
2. *Familiarisation with the transcripts* by the main researcher (CDL), who will write down analytical notes on margins.
3. *Coding of a sample of three transcripts* by the main researcher, a second researcher within the research team and one PPI contributor, who will independently underline relevant pieces of text and write coding labels for each, reflecting the constructs included in the topic guide. However, to prevent the omission of important data, if novel constructs are identified from the transcripts, new coding labels will be generated.

4. *Development of a working analytical framework* through teamwork of the three coders, who will create a set of initial codes through the synthesis of individual coding and operational definitions. Two more transcripts will be coded by two coders to check whether the initial working analytical framework is suitable. Eventually, a stable set of codes, clustered into umbrella categories will be identified.
5. *Use of the working analytical framework* by the main researcher (CDL) to code the whole set of transcripts in NVivo V.12.[45] Double coding will be conducted by another researcher.
6. *Charting of data into the framework matrix* by the main researcher on NVivo. The matrix will map out codes (one per column) and participants (one per row). The relevant quotes will be transferred from NVivo onto the matrix.
7. *Interpretation of data* by the main researcher, who will develop themes from the matrix by making connections within and between participants and categories. This will be an iterative process, with regular review from members of the research team.

## ETHICS AND DISSEMINATION

The PrAISED trial and process evaluation have received ethical approval number 18/YH/0059.

This protocol, grounded in the Medical Research Council (MRC) framework for process evaluation of complex intervention,[40] outlines the rationale, design and methods for the process evaluation of the PrAISED and mild cognitive impairment, following the changes made as a result of the restrictions on face-to-face contact during the COVID-19 pandemic.

In only a few months, the COVID-19 pandemic has required dramatic changes to our lifestyles and caused unprecedented operational strain on national health and social care systems. There is a need for scientific evidence to inform research and services in response to the current challenges, as well as preparation for services after the pandemic and potential future events. In these respects, the final process evaluation report, which will be disseminated in scientific journals and to the public (eg, through public engagement events), will report on the impact that the social distancing measures introduced in PrAISED have had on research participants and therapists. By comparing the evidence gathered through this study with the original PrAISED process evaluation[25] and the wider literature, this process evaluation will contribute knowledge on ways in which individuals belonging to the most vulnerable groups in society can be better supported and motivated to remain physically active and healthy in their homes without face-to-face support. In addition, by triangulating data from this process evaluation with some quantitative measures from the RCT (eg, QoL and carer strain), we will be able to gather a more comprehensive picture of the impact that the COVID-19 has had on the lives of participants.

This work will also present important implications in theory advancement. Our dissemination plans include a paper further validating the *PHYT in dementia*, the behaviour change theoretical model that our research team previously developed and validated through data from the original PrAISED process evaluation.[26][27] Results from this work will contribute further evidence to confirm/challenge the validity of the model in explaining motivation to be physically active, in the context of social distancing. Finally, based on findings from this process evaluation, we aim to develop a methodological paper outlining strategies that can be used to involve research participants remotely in an ethical, meaningful and practically feasible way. This model can be refined through input from research teams conducting rehabilitation studies in similar circumstances, such as the FinCH study,[46] to derive a research platform that can be shared to inform/guide good practice in future research.

In conclusion, this process evaluation represents one of the first efforts to document how an ongoing research programme was adapted as a result of the COVID-19 pandemic. This study will support the critical reflection by the PrAISED team on positive and negative aspects of these adaptations. It will also provide transferable information to develop strategies to effectively deliver rehabilitation remotely, in the presence of extraordinary circumstances (eg, social distancing and staying at home).

**Contributors** CDL contributed to the conception and design of the study, and the development of all the elements of the process evaluation. AB contributed to the planning of the process evaluation, the analysis plan for therapists' audio recordings and provided feedback and final approval of the manuscript. SEG contributed the PrAISED RCT information, helped in the conception of the study and provided feedback and final approval of the manuscript. RN contributed to the development of the analysis plan for the qualitative element of the study and the therapists' audio recordings and provided feedback and final approval of the manuscript. ROB contributed to develop the study design and the analysis plan of the therapists' audio recordings and provided feedback and final approval of the manuscript. LH helped to develop the topic guide for the therapists' qualitative interviews, as well as the analysis plan and provided feedback and final approval of the manuscript. VvdW contributed to the study conception and design, the implementation study, the quantitative data analysis plan and provided feedback and final approval of the manuscript. KP provided guidance and expertise on the development of the qualitative interviews for participants with dementia and carers and provided feedback and final approval of the manuscript. VB contributed to the design and analysis plan of the implementation study and provided feedback and final approval of the manuscript. PL contributed to the discussion section of the manuscript and provided feedback and final approval of the manuscript. MG and MD were the PPI collaborators of the study. They contributed to the development of the topic guide for the qualitative interviews of the participants with dementia and their carers and provided feedback and final approval of the manuscript. JH contributed to the discussion section of the manuscript and provided feedback and final approval of the manuscript. RHH contributed the PrAISED RCT information, helped in the conception of the study and provided feedback and final approval of the manuscript.

**Funding** This protocol presents independent research funded by the United Kingdom National Institute for Health Research (NIHR) under its Programme Grants for Applied Research funding scheme (RP-PG-0614-20007). The views expressed are those of the authors and not necessarily those of the National Health Service, the NIHR or the Department of Health and Social Care.

**Competing interests** None declared.

**Patient and public involvement** Patients and/or the public were involved in the design, or conduct, or reporting, or dissemination plans of this research. Refer to the Methods section for further details.

**Patient consent for publication** Not required.

**Provenance and peer review** Not commissioned; externally peer reviewed.

**ORCID iD**
Claudio Di Lorito http://orcid.org/0000-0002-8953-0117

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
