## [Reviewer comments · BMJ Open]

ARTICLE DETAILS

TITLE (PROVISIONAL)	Protocol for the process evaluation of the Promoting Activity, Independence and stability in early Dementia (PrAISED), following changes required by the COVID-19 pandemic
AUTHORS	Di Lorito, Claudio; Bosco, Alessandro; Goldberg, Sarah; Nair, Roshan; O'Brien, Rebecca; Howe, Louise; van der Wardt, Veronika; Pollock, Kristian; Booth, Vicky; Logan, Phillipa; Godfrey, Maureen; Dunlop, Marianne; Horne, Jane; Harwood, Rowan

VERSION 1 – REVIEW

REVIEWER	Gabriele Cipriani Versilia Hospital, Neurology Unit. Italy
REVIEW RETURNED	28-Apr-2020

GENERAL COMMENTS	Reviewer has no comments.
---------------------------

REVIEWER	Rachel Potter University of Warwick, UK
REVIEW RETURNED	12-May-2020

GENERAL COMMENTS	This will be a useful piece of work evaluating the impact of COVID-19 on trial processes and implementation of the study intervention. Understanding if, and how, it may be possible to support people with dementia remotely to increase physical activity will be an interesting finding. I found it difficult at times to know where you are up to in the study, and what still needs to be achieved; this could be made clearer at the beginning of the manuscript. Paragraph 2 of the Introduction reads as if the trial is complete. I would make it clear to the reader how many participants all likely to be affected by the changes to protocol. Page 4 line 12 reference to the appendix needs adding. I also found the premise for the implementation study unclear. it appears that all the therapist training was complete before the challenges of COVID-19, (Table 1) ? so I'm not sure why hours of training/attendance of training and completion rates of assessment questionnaires are relevant? (page 6). The rationale for this needs to be clarified. Exactly how has the training been changed post COVID-19? to answer the the research question 'How are therapists trained to deliver a physical activity and exercise programme remotely to participants with dementia' being addressed? For the study on mechanisms of impact and context it would be useful to have an idea of how many participants from each group will be interviewed. Some of the dissemination section (from line 45 page 10 to line 46 page 11) would perhaps sit better in the Introduction? It isn't
---

	dissemination. It could also be condensed.
REVIEWER	YU, Xin Peking University, Institute of Mental Health China
REVIEW RETURNED	13-May-2020
GENERAL COMMENTS	 1. The possible duration of the study should be described. 2. The situation and the government policy of COVID-19 pandemic is changing, and the influence of the change on the adaption of intervention should be considered 3. Besides interview, whether using some tests or scales to evaluate the effect of the intervention change on the patients' quality of life would be necessary ? 4. About the Implementation study, were patients in the control arm included? The descriptions in figure 1 and the text in page 5 line 37-38 were different. 5. About the procedure of the mechanisms of impact study shown in figure 1, the control arm should receive phone interview after change of intervention while they do not receive intervention. It seems confusing. 6. The sample size in each study should be stated. 7. About the mechanisms of impact study, the detail of purposive sampling should be provided (such as how to select participants with dementia in the each arm).

VERSION 1 – AUTHOR RESPONSE

Reviewer 2	I found it difficult at times to know where you are up to in the study, and what still needs to be achieved; this could be made clearer at the beginning of the manuscript. Paragraph 2 of the Introduction reads as if the trial is complete. I would make it clear to the reader how many participants all likely to be affected by the changes to protocol.	We have clarified this in the introduction, as per reviewer's comment	Introduction
	Page 4 line 12 reference to the appendix needs adding.	We have added a reference to the Appendix	Introduction
	I also found the premise for the implementation study unclear. It appears that all the therapist training was complete before the challenges of COVID-19, (Table 1)? So I'm not sure why hours of training/attendance of training and completion rates of assessment questionnaires are relevant? (Page 6). The rationale for this needs to be clarified. Exactly how has the training been changed post COVID-19? To answer the research question 'How are therapists trained to deliver a	In recognition of the validity of the reviewer's comment, we have taken out the training element from the fidelity study, as this was not changed as a result of COVID-19. We will now only look at delivery, which has changed as a result of COVID-19.	Study research questions, implementation study section and figure 1

	physical activity and exercise programme remotely to participants with dementia' being addressed?		
	For the study on mechanisms of impact and context it would be useful to have an idea of how many participants from each group will be interviewed.	We have added this information	The study of mechanisms of impact and context participants' section
	Some of the dissemination section (from line 45 page 10 to line 46 page 11) would perhaps sit better in the Introduction? It isn't dissemination. It could also be condensed.	We agree. We have condensed the section in question and moved it in the introduction	Introduction
Reviewer 3	The possible duration of the study should be described.	We have added this information	Methods
	The situation and the government policy of COVID-19 pandemic is changing, and the influence of the change on the adaptation of intervention should be considered	We agree that the situation is ever changing. Our study will be able to capture through the implementation study ("adaptations" section) and the qualitative interviews (carried out at different time points) these changes and the impact they will have on the study participants	
	Besides interview, whether using some tests or scales to evaluate the effect of the intervention change on the patients' quality of life would be necessary?	We agree with this comment. The PrAISED trial, which is ongoing, will be able to gather these data from study participants through QoL assessments. There is a possibility to triangulate these data with data gathered on QoL through the process evaluation (e.g. qualitative interviews), to gather a more comprehensive picture of the impact of COVID-10 on QoL. We have now added this as a potential dissemination output in the ethics and dissemination section	Ethics and dissemination section
	About the Implementation study, were patients in the control arm included? The descriptions in figure 1 and the text in page 5 line 37-38 were different.	We just include participants in the intervention arm. We have corrected the mistake in Figure 1	Figure 1
	About the procedure of the mechanisms of impact study shown in figure 1, the control arm should receive phone interview after change of intervention while they do not receive intervention. It seems	They receive the phone interview anyway, because we want to investigate whether there are any relevant differences between the controls and	Figure 1 and study of mechanisms of impact and context participants'

	confusing.	the participants who receive the phone support. We have modified figure 1 and added this information in the participants' section to clarify	section
	The sample size in each study should be stated.	For the qualitative study, we cannot clearly define a sample size a priori, as we are using conceptual density. However, we have added an expectations of the sample size.	The study of mechanisms of impact and context participants' section
	About the mechanisms of impact study, the detail of purposive sampling should be provided (such as how to select participants with dementia in the each arm).	We have now explained this	The study of mechanisms of impact and context participants' section

VERSION 2 – REVIEW

REVIEWER	Rachel Potter University of Warwick UK
REVIEW RETURNED	09-Jun-2020

GENERAL COMMENTS	A couple of small things that could do with revising before publication, but that don't stop me recommending for publication: Page 3 line 24, suggest adding the recruitment target Page 4 line 17, remover 'has issued' Page 4, line 27 'may incur into a lack of social contacts' needs rephrasing All the very best with the study.
--

VERSION 2 – AUTHOR RESPONSE

Comment	How we addressed the comment	Where we addressed the comment
Page 3 line 24, suggest adding the recruitment target	We have now added the recruitment target	Introduction, page 3
Page 4 line 17, remover 'has issued'	We have now removed this	Introduction, page 4
Page 4, line 27 'may incur into a lack of social contacts' needs rephrasing	We have rephrased to make it clearer	Introduction, page 4